# Gluten-Free Cereal Products and Beverages: A Review of Their Health Benefits in the Last Five Years

**DOI:** 10.3390/foods10112523

**Published:** 2021-10-21

**Authors:** Muhammad Arif Najmi Khairuddin, Ola Lasekan

**Affiliations:** Department of Food Technology, University Putra Malaysia UPM, Serdang 43400, Malaysia; arifnajmi.khairuddin@gmail.com

**Keywords:** gluten-free products, cereals, pseudo-cereals, gluten-free beverages, health benefits

## Abstract

In the past decades, food products and beverages made from gluten-free cereals were initially created for certain groups of people who experience gluten-related disorders such as wheat allergies, gluten ataxia, non-celiac gluten sensitivity, and the most well-known, celiac disease. Nowadays, the consumption of gluten-free products is not only restricted to targeted groups, but it has become a food trend for normal consumers, especially in countries such as the UK, the US, and some European countries, who believe that consuming a gluten-free product is a healthier choice compared to normal gluten-containing products. However, some research studies have disapproved of this claim because the currently available gluten-free products in the market are generally known to be lower in proteins, vitamins, and minerals and to contain higher lipids, sugar, and salt compared to their gluten-containing counterparts. The use of other gluten-free cereals such as sorghum, millet, and teff as well as pseudo cereals such as buckwheat and quinoa has gained significant interest in research in terms of their various potential health benefits. Hence, this review highlights the potential health benefits of some gluten-free cereals and pseudo cereals apart from corn and rice in the last decade. The potential health benefits of gluten-free products such as bread, pasta, crackers, and cookies and the health benefits of some other non-alcoholic beverages made from gluten-free cereals and pseudo cereals are reported.

## 1. Introduction

The term gluten-free in food products is usually associated with a certain group of the world’s population that experience gluten-related disorders or diseases. In other words, gluten-free food products are initially made for consumers who suffer from gluten-related diseases such as wheat allergies, gluten ataxia, non-celiac gluten sensitivity, wheat-dependent, exercise-induced anaphylaxis (WDEIA), and the most well-known, celiac disease. Celiac disease is an immune-mediated enteropathy experienced by individuals who are genetically susceptible to gluten exposure in their diet [1]. Celiac disease is known to affect 1% of the world’s population [2], and strict gluten-free diets need to be implemented by these individuals in order to experience relief. Non-celiac gluten sensitivity (NCGS) is a clinical entity caused by the consumption of gluten-containing products which results in intestinal or extra intestinal symptoms or signs, or even both [3]. Usually, a significant improvement in symptoms is achieved once the gluten-containing product is removed from the diet of the patient [1]. However, this definition has been criticized in recent reports [3,4]. For instance, the causation or etiology of NCGS as well as its link to irritable bowel syndrome raises some fundamental issues. Sergi et al. [4] opined that NCGS is not the only cause of irritable bowel syndrome and the authors affirmed that other components of wheat such as non-protein fractions could trigger the symptoms in patients with this functional disorder. For instance, wheat amylase and trypsin-inhibitors, complex proteins which trigger innate immunity, could also stimulate the symptoms of NCGS [5]. Similarly, the authors also reported that fermentable oligo-saccharides, di-saccharides, mono-saccharides, and polyols (FODMAPs) may play a role in producing gastrointestinal symptoms in patients with NCGS [5]. Moreover, the avoidance of a diet low in FODMAPs results in an improvement of the clinical picture of patients [5]. All these reports support the role of these dietary factors in symptoms of NCGS rather than gluten. Gluten ataxia was originally defined as idiopathic irregular ataxia with the presence of anti-gliadin antibodies of the IgG and/or IgA type and it was considered as one of the most common neurological manifestations of a gluten-free disorder. Meanwhile, wheat allergy is another immunological reaction that occurs due to the presence of protein inside wheat and other related grains, and depends on the exposure routes. They often manifest different clinical symptoms [1]. On the other hand, WDEIA is a severe variant of a food allergy caused by physical exercise or by other co-factors in combination with the ingestion of food containing wheat [6]. Studies have shown that the clinical threshold in WDEIA seems to be lowered in patients on a wheat free diet while it is noticeably high in those with a regular wheat intake [6].

Studies have shown that the only known option to minimize the negative effects of consuming gluten-containing products is to avoid or completely eliminate these products from the diet [7]. In 1941, the first gluten-free diet made its first debut [8]. Today, the term gluten-free diet is considered common, especially in the US and Europe, and extensive research has been performed to create food products that are gluten-free. However, today’s current trend shows that gluten-free products are not only consumed by individuals with gluten-related disorders, but are also preferred and consumed by those without any symptoms related to gluten disorders. As reported by Khoury et al. [8] the reasons behind this phenomenon are mainly sourced from their beliefs that gluten-free products are healthier, or the avoidance of gluten products would promote their general health. In fact, another study by Gobbetti et al. [7] also mentioned that people who are gluten sensitive or who have family members with gluten sensitivity represent the consumers most likely to buy gluten-free products.

Cereal-based foods are considered a staple for people all across the globe. The sales of gluten-free products as presented by the global market data are predicted to increase by 7.6% of its compound annual growth rate from 2020 to 2027 [9]. Gluten-free cereals are cereals without the presence of gluten, or which contain less than 20 parts per million (ppm) of gluten in the cereal [8] and examples of these cereals include maize, rice, sorghum, teff, and millet. Oats can be considered as a gluten-free cereal product as reported by several studies [8,9,10]. However, there are some issues regarding the claims that oats are a gluten-free cereal; oats are described as one of the gluten containing cereals by the Codex Alimentarius [11] and the presence of symptoms of exacerbation and malabsorption are believed to be connected to the inclusion of oats in gluten-free diets [11]. There are beverages made from cereals that are gluten-free. Examples of gluten-free beverages include beverages that are derived from gluten-free cereals such as millet, teff, and sorghum [12].

There is a rising concern that people who consume the market-available gluten-free products are likely to experience some health-related problems such as obesity, cardiovascular risks, and metabolic syndrome [13]. There is a need to further address and evaluate the potential health benefits of gluten-free products such as food and beverage products derived from gluten-free cereals, as well as from pseudo cereals. The objective of this review is to present a review of the potential health benefits of gluten-free cereal products and beverages in the last five years. Besides that, the controversial issues regarding the consumption of gluten-free products are also discussed in this paper.

## 2. Potential Health Benefits of Gluten-Free Cereal Products and Beverages

The avoidance of gluten-containing products by consumers who are susceptible to gluten-related disorders is inevitable. Gluten-related disorders, particularly celiac disease, are triggered by the ingestion of a storage protein known as gluten that is naturally present in grains such as rye, wheat, and barley. However, the consumption of gluten-free products which are available in the market has been shown to produce no significant improvement in the conditions of consumers with a gluten-related disorder. Research by Gobbetti et al. [7] showed that gluten-free products targeted at children were not superior to common food items targeted at children in terms of their nutritional content. A recent comparison between the number of nutrient inadequacies in celiac disease patients at the point of diagnosis and celiac patients who adhered to a gluten-free diet showed a greater number of nutrient inadequacies in the second group [14]. Additionally, other studies have also reported that people who practice a gluten-free diet often experience nutrient imbalances. For example, some of the people that regularly consume gluten-free cereal-based products are known to have a higher sugar and fat intake which could lead to the risk of obesity and other health problems [7]. Cereals that are gluten-free such as maize, rice, sorghum, millet, and teff, as well as pseudo cereals such as fonio, buckwheat, and quinoa possess many potential health benefits that can be used to improve the nutritional weaknesses of the current commercially available gluten-free products.

### 2.1. Health Benefits of Some Gluten-Free Cereals and Pseudo Cereals

#### 2.1.1. Sorghum

Sorghum is known to be the fifth most largely cultivated cereal crop in the world and is the major source of food for humans in some African countries where agricultural activity and the surrounding environment for planting other crops is unsuitable [12]. The use of sorghum for human food has gained significant interest as it has a high nutrient potential and can be safely consumed by people with celiac disease [12]. Sorghum consists of three major parts which are the bran layer, the endosperm, and the germ. Sorghum is an abundant fiber source as it is a non-starch carbohydrate which is comprised of insoluble fibers and soluble fibers that are located both in the endosperm cell walls and the pericarp and contribute about 6–15% of the grain’s total weight [15]. Sorghum is also an excellent food for people with diabetes and obesity issues. It is characterized as having a low rate of protein digestibility and a low starch content [15]. Other than that, good antioxidant properties are also associated with sorghum grains. Research completed by Roa et al. [16] showed that sorghum contained greater antioxidant activities than millet, and the authors further reported that the sorghum variety which had a higher antioxidant activity had a greater quantity and diversity of poly phenol (Table 1). The function of antioxidant activity in the body is the neutralization and reduction in the reactive oxygen species that are involved in the process formation of many varieties of diseases such as cardiovascular disease, cancer, aging, and neuro- degenerative diseases.

#### 2.1.2. Millet

Compared to other main cereals, millet cereal is considered as a type of crop which is resistant to drought, disease, and pests and has a short growing season [22]. Millet is an important crop in Africa and Asia with 97% of its production in developing countries and India is its the largest producer [22]. Several studies have reported the health and nutritional benefits of several varieties of millet. For instance, fox-tail millet has been known to show potential to lower the risk of chronic diseases such as type 2 diabetes mellitus and cholesterol metabolism when fox-tail millet is included in a normal routine diet [22,23]. Pearl millet contains properties that could promote health such as correcting the hyperglycemia caused by type 2 diabetes, the possession of anti-cancer properties, and the promotion of pro-biotic growth [24]. Besides that, the nutritional composition of finger millet has been known to reduce the risk of high blood pressure, diabetes mellitus, and the disorder related to gastrointestinal tract when the grains are absorbed into the body [25]. That being said, millet cereal contains health benefits that include anti-diabetic and anti-carcinogenic properties and an ability to reduce constipation problems (Table 1) as well as other potential benefits to human health that were not mentioned earlier, such as the ability to enhance hemoglobin levels [25].

#### 2.1.3. Teff

Teff, also known as one of the grains from the Poaceae grass family, is an annual crop indigenous to the Ethiopians and is believed to have the smallest grain size in the whole world [12,19]. Teff is mainly cultivated for its small seeds or grains which are used in making a staple food known as injera, a spongy flatbread made from fermented flour, as well as in the making of a traditional porridge and an alcoholic beverage [12,19]. Teff contains a high concentration of insoluble polysaccharides and macro- and micro-nutrients [19] (Table 1). A brown-colored variety of teff has gained the interest of health-conscious Ethiopians for the past several years as the variety is believed to have higher nutritional values [12]. Teff cereals are suitable for people with type 2 diabetes as they have a low glycemic index and are naturally rich in iron which makes it a suitable food for consumers with anemia [19]. Besides that, teff is also reported to have higher minerals such as zinc, iron, and calcium compared to other grains [26]. Moreover, teff also displays antioxidant properties and is effective in the treatment of diabetes and anemia [19] (Table 1). Barretto et al. [26] reported that ferulic acid, a phenolic acid compound which is one of the most vital antioxidant agents, is the major compound found in teff. Other bio-active components of this grain are summarized in Table 1.

#### 2.1.4. Buckwheat

Buckwheat is a dicotyledonous pseudo cereal that belongs to the Polygonaceae family. Buckwheat is mostly grown in the northern region of the globe, and the world’s main producers of buckwheat are China and Russia [27]. Buckwheat is normally consumed in seed form [27]. It is rich in carbohydrates, proteins, amino acids, fatty acids, vitamins, minerals, and iminosugars (Table 1) [20]. The two most commonly consumed buckwheat varieties are common buckwheat and Tartary buckwheat [27]. Similar to sorghum, millet and teff, buckwheat is also gluten-free and at the same time, possesses many health and nutritional benefits. Buckwheat has been extensively researched and is frequently associated with health benefits including antioxidant, anti-cancer, anti-inflammatory and anti-hypertension properties [27]. Their antioxidant activity is attributed to the presence of various bio-active compounds such as rutin, quercetin, orientin, isoorientin, and isovitexin (Table 1) [20]. For instance, Giménez-Bastida and Zieliński [27] reported that tartary buckwheat possesses higher levels of the major polyphenol (rutin) than common buckwheat and this might be one of the factors that confers a higher antioxidant activity on Tartary buckwheat than the common buckwheat.

#### 2.1.5. Quinoa

Similar to buckwheat, quinoa is also a pseudo cereal belonging to the Chenopodiaceae family. Quinoa is believed to have originated from the Andean region, particularly from Peru and Bolivia. The grain is rich in protein with values ranging from 14 to 18% [21] (Table 1). It has a balanced amino acid composition and vitamins. Quinoa has been reported to possess a wide range of health benefits such as antioxidant, anti-obesity, and hypocholesterolemic effects [28]. In a previous study conducted by Tang and Tsoa [29], a higher antioxidant activity and the presence of phenolic compounds in quinoa seeds with darker color were observed. Recently, Teng et al. [28] reported that the bio-active polysaccharide that was extracted from quinoa showed an ability to suppress lipid accumulation. Besides that, an earlier study by Vilcacundo and Hernández-Ledesma [30] reported that phytochemicals commonly found in quinoa such as phytosterol could reduce cholesterol serum levels by competing for intestinal absorption with the cholesterol. Health benefits related to antioxidant activity and gastrointestinal and cardiovascular health have been reported [29]. Furthermore, a summary of its nutrients and bio-active compounds is listed in Table 1.

### 2.2. Health Benefits of Gluten-Free Cereal Products and Beverages

Gluten-containing products are products which are manufactured from gluten-containing cereals such as rye, wheat, oats, and barley. Grains other than wheat are known to contain ‘gluten’ as their storage protein even though they exist and can be referred to as hordein, avenins, and secalin which are contained in barley, oats, and rye, respectively [31]. Since some gluten-containing products such as pasta and bread are considered as a staple for major parts of the world’s population, food innovated from gluten-free cereals is a must, especially for those who have adverse reactions to gluten, such as those with gluten-sensitivity and celiac disease. Some studies have suggested that gluten-free products which are available in the market are inferior to the gluten-containing products in terms of nutritional quality [32]. Therefore, some highlights on some of the potential health benefits of gluten-free cereal-based products such as bread, pasta, and other types of products such as crackers and cookies, as well as gluten-free beverages, are shown in (Table 2).

#### 2.2.1. Health Benefits of Gluten-Free Bread, Pasta, Cookies, and Crackers

Bread is one of the major staple foods for people in the world. The main reason for this is its main nutritional composition (carbohydrate). Several studies have been conducted to determine the potential health benefits of bread made from gluten-free cereals. An example of this includes bread made from sorghum [33]. The authors reported that the use of sorghum flour in making Chinese steamed bread increased the amount of the total phenolic content, which resulted in increased antioxidant activity, especially when red sorghum flour was used compared to white sorghum flour [33]. In another study, Girard and Awika [48] reported that bread made with sorghum flour had a very low glycemic index which contributes to prolonged satiety. Similarly, the authors showed that the glycemic indices as well as the satiety levels of breads made from other flours increased in the following order: buckwheat, quinoa, and wheat. Besides that, bread made from millet is also believed to have an anti-hypertension effect. According to Hou et al. [34], milled steam breads which were made from a pure 50 g of foxtail millet flour were given to healthy participants with mild hypertension and the results showed a significant reduction in blood pressure among the participants. In addition, millet bread also exhibited a glucose-lowering effect [34] in participants with an impaired glucose intolerance. The reason behind the anti-hypertension might be the protein hydrolysate found in foxtail millet which has the capacity to reduce blood pressure [34]. A report by Hou et al. [34] showed that the addition of other cereals such as teff flour into bread could increase its protein, dietary fiber, and mineral contents. Apart from that, pseudo cereals such as buckwheat and quinoa also have potential health benefits when used in making bread. In research conducted by Verardo et al. [35] the authors revealed that bread enriched with buckwheat flour had an increased content of flavan-3-ols and flavanols and exhibited a corresponding increase in radical scavenging activity. In another study, black, white, and red varieties of quinoa flours were incorporated into bread by Ballester-Sánchez et al. [36] and the authors observed that the black and red quinoa flours were the most effective in raising the antioxidant properties in the bread.

Pasta is traditionally made from the semolina (coarse flour) of durum wheat. It is also considered as a famous and recognized food for people around the world. Research completed by Palavecino et al. [49] reported that brown and white sorghum-based pasta showed a high nutritional content of dietary fiber, protein, and polyphenols as well as having high health benefits in terms of its antioxidant activity. Other health benefits of pasta made from sorghum flour included a low glycemic index, a high resistant starch that contributed to a reduced rate of sugar absorption, LDL-cholesterol and polyphenol, which helps to lower the risk of several cancers [37]. Pastas made from finger millet and sorghum flours have also been reported to show greater antioxidant activity potential compared to a control sample of durum wheat pasta. Besides that, the use of pseudo cereals such as quinoa, buckwheat, and amaranth without the addition of eggs that are normally used in pasta making were found to be good sources of dietary fiber, protein, minerals, and phytonutrients [38]. Other products such as crackers and cookies have also been studied in order to create a gluten-free version of these products. As reviewed by Martínez-Villaluenga et al. [39], gluten-free cereal flour from quinoa grains was demonstrated to have high dietary fiber, antioxidant activity, and essential amino acids when used as ingredient in cookies. Research by Molinari et al. [40] found that Tartary Malt Cookies from germinated Tartary buckwheat contained a higher antioxidant capacity and a low glycemic index and phenolic content compared to the control rice cookies and Tartary flour cookies. Crackers made with the addition of buckwheat have been reported to exhibit a higher antioxidant activity than the control samples made from wheat crackers, while sourdough crackers had the highest antioxidant activity among the three crackers. These findings were corroborated by those of Xu et al. [41], which stated that buckwheat crackers made from refined or wholegrain buckwheat flour showed a higher antioxidant activity than crackers made from wheat.

#### 2.2.2. Health Benefits of Gluten-Free Beverages

Fermented cereal beverages were originally produced primarily for food preservation, health benefits, and flavor enhancement [50]. Due to the increased popularity of gluten-free products, many beverage companies are moving towards that market opportunity. Gluten-free beverages can be divided into two categories: gluten-free alcoholic beverages and gluten-free non-alcoholic beverages. The most common gluten-containing alcoholic beverage is beer. Beer is considered to be an ancient alcoholic drink and it has positioned itself as the third most famous alcoholic drink in the world prior to water and tea. Some studies have reported on the production of gluten-free beer with a regular beer flavor to cater for gluten-sensitive consumers [51]. Beverages are known to deliver bio-active components, nutrients, minerals, antioxidants, vitamins, fatty acids, probiotics, prebiotics, and micro-nutrients to consumers [51]. The combination of cereals with pseudo cereals such as quinoa, amaranth, and others have been reported to intensify the availability of beverages rich in bio-active compounds [50]. Presently, there are commercially available gluten-free non-alcoholic beverages such as Amake, Boza, and Borş [50].

Several studies have reported on the health-promoting benefits of gluten-free beverages. For instance, Kaur and Tanwar [42] reported that beverages made from malted quinoa had the potential for anti-hypertensive and anti-diabetic properties, respectively. Besides that, Queiroz et al. [47] also demonstrated that a powdered drink mix made from sorghum flour which contained tannin showed significant antioxidant activity with a low calorific value. Another study by Ajiboye et al. [43] on the effect of a non-alcoholic beverage (obiolor) produced from fermented sorghum and millet malts on the dyslipidemia, protein oxidation, lipid peroxidation, and DNA fragmentation in the liver of rats revealed a significant reduction in the DNA fragmentation, the peroxidation of lipids, protein oxidation, and dyslipidemia in the experimental rats. In addition, the production of a beverage such as tea from the black-colored sorghum variety with a high phenolic content showed a higher antioxidant activity compared to the white-colored variety of sorghum grains [44].

Apart from that, the beneficial health effects of these gluten-free beverages could also be enhanced by incorporating certain strains of microbial cultures or adding other natural ingredients. For example, Badejo et al. [45] demonstrated that fortifying a local African non-alcoholic cereal-based beverage, known as Kunnu with a tiger nut extract significantly enhanced its antioxidative potential without changing its good overall acceptability. Other than that, the addition of microbial strains was also observed to improve the nutrient contents of the gluten-free cereal beverages via fermentation. This was shown in a study by Adeyanju et al. [46], in which a non-alcoholic beverage made from sorghum and amaranth with the addition of lactic acid bacteria for souring through lactic acid fermentation displayed a positive outcome. In their results, the bio-accessibility of zinc was elevated from 24 to 194%, while the bio-accessibility of iron had escalated from 128 to 372% compared to the control beverage with no acidification or fermentation process [46]. Other results also showed incremental changes in the digestibility of protein, bio-accessible iron, and available lysine by 14–58%, 34–64% and 24–52%, respectively, in the beverage compared to the beverage which was produced with 100 % sorghum [46]. This was similar to the findings of Ogodo et al. [52] which stated that the drop in pH in lactic acid bacteria-fermented sorghum flour showed an improvement in protein digestibility and a favorable condition for the peptidase enzyme activity.

## 3. Issues Related to the Consumption of Gluten-Free Products

### 3.1. Nutrient Insufficiency

One of the most worrisome issues regarding the consumption of gluten-free products is the nutrient insufficiency in most commercialized gluten-free products that are currently available in the market. For instance, comparative research by Jamieson et al. [53] on Canadian gluten-free food products found that staple gluten-free foods such as pasta, bread, flour, and cereals contained a 1.3% higher fat content with less protein, iron, and folate compared to their counterpart products that contained gluten. Based on Fry, Madden and Fallaize [54], who conducted research on the nutrient composition of gluten-containing and gluten-free products across the UK, found that medium and high levels of fat, as well as saturated fat, salt, and sugar were often observed in gluten-free food products; however, the results were inconsistent as gluten-containing products such as flour and bread had more sugar and fat, while gluten-free crackers had higher levels of sugar and fat than the regular gluten-containing crackers. They also found that a high content of salt was also commonly found in gluten-free products available in the UK supermarket [54]. Another study by Babio et al. [55] who carried out research in the Spanish market also discovered that some gluten-free products containing a higher amount of sugar and fat with lower amounts of protein and dietary fiber were sold at higher prices compared to those products that were not gluten-free. This was similar to the observation of Fry et al. [54] on the Australian market. The authors reported that most gluten-free food products had lower average protein contents than gluten-containing food, especially bread and pasta. In addition, they found that the gluten-free products were not superior to the products that contained gluten in terms of nutritional quality. Besides that, gluten-free products were also associated with poor micro-nutrients such as B vitamins and minerals such as calcium, zinc, iron, and magnesium [56]. Therefore, nutrient imbalances that are generally found in gluten-free products that are available in the market might lead to potential health risks. An article by Niland and Cash [57] listed some of the potential health risks that non-celiac people could be exposed to if they continued to consume commercialized gluten-free products, and these included hyperglycemia, coronary disease, hyperlipidemia, and deficiencies in dietary fiber and micro-nutrient contents. They also highlighted the increased financial burden due to the high cost of most commercialized gluten-free foods.

A possible reason contributing to the imbalanced nutrient content of most commercialized gluten-free products is probably the ingredients used. Most of the raw ingredients are sourced from flour or/and starch originating from corn and rice [9]. These raw ingredients (corn and rice flours) are generally low in lipids, ash, and crude fiber. For instance, Altindag et al. [58] conducted a comparative proximate analysis on some of these cereals and showed that rice flour contained the lowest content of lipids, ash, crude fiber, and moisture while corn flour had the highest crude fiber and lipid content, but had the lowest protein content. Meanwhile, the authors reported that buckwheat flour was significantly higher in protein and ash content among the three flours. In addition, some of the gluten-free products were also attributed a high glycemic index value. Molinari et al. [40] observed that raw rice flour showed the highest expected glycemic index value compared to the whole Tartary buckwheat malt and the whole Tartary buckwheat flour. This can be explained by the presence of lower resistant starch and dietary fiber in rice which probably accounts for the high rate of starch hydrolysis, and its high expected glycemic index value [40]. Furthermore, the deficiency in micro-nutrients found in some gluten-containing products is due to several reasons. For example, Pellegrini and Agostoni [59] opined that the presence of low micro-nutrients such as niacin, thiamine, calcium, and iron in gluten-free products was mainly due to the fact that gluten-free products were not fortified or enriched with these vitamins and minerals as compared to gluten-containing products; B vitamins and minerals fortification are mandatory for gluten-containing products such as wheat flour in countries such as the UK and US.

Despite the removal of the offending allergen from the diet of patients with NCGS through the adoption of a gluten-free diet, mineral disorders still constitute a serious problem [60]. Most patients experience deficiencies in calcium, magnesium, iron, and zinc [60]. In recent research on the enrichment of gluten-free breads made from buckwheat flour and rice flour and fortified with milk and different seeds (i.e., flax, poppy, sunflower seeds, pumpkin seeds, or nuts) revealed a higher bioavailability of calcium, magnesium and iron in the rice bread, while the buckwheat bread produced more bioavailability of zinc and copper [60]. This finding is a clear indication of the need for new products which will address the needs of patients requiring nutrient supplementation.

### 3.2. Undesirable Quality of Products

The presence of gluten in bakery products is considered necessary especially in bread products for the formation of the necessary networks that entrap bubbles of air needed to create a bread with a desirable large volume [61]. The two protein fractions of gluten, glutenin, and gliadin are widely believed to be responsible for the elasticity of the network inside the wheat flour dough and the increase in the dough’s viscosity, respectively [8]. Thus, the exclusion of gluten in bread-making would lead to undesirable properties such as insufficient gas retention which would later result in a low loaf volume and the formation of liquid batter rather than dough that usually contributes to poor color development, texture crumbling, and other quality defects after baking [62]. However, this does not seem to be an issue in bakery products such as cookies and cakes, since the gluten network-forming character is not considered as important in these types of products; however, they are still believed to have an effect on the final texture and structure of the cookies and cakes [8].

Besides that, the sensory properties of commercialized gluten-free products are not at par with the gluten-containing products. Naqash et al. [62] observed that it might be quite difficult to impart a similar desirable taste, aroma, and texture in breads that are gluten-free with those that contain gluten. This was proven by research conducted by Kaur et al. [63] on biscuits made from buckwheat and wheat flours. Buckwheat biscuits produced the lowest values in taste, color, flavor, appearance, and overall acceptability compared to wheat flour biscuits and biscuits from buckwheat flour with added edible gums [63].

### 3.3. Adulteration of Gluten-Free Products with Gluten

Gluten contamination is also one of the main concerns when buying products that are declared gluten-free. Some studies have shown that commercialized gluten-free food products are often not gluten free. For instance, Farage et al. [64] sampled a total of 130 gluten-free products such as bread, biscuits, and cakes from bakeries across Brazil and the authors found that 28 samples had gluten content which exceeded 20 ppm. However, the result varies between different countries, as shown by Verma et al. [65] who found that out of 173 products labeled as naturally gluten-free, only 9% of the products recorded 20 ppm of gluten content. Apart from that, gluten-free products made from cereals such as oats are of some concern since they are prone to gluten contamination. Essentially, oats are considered as gluten-free since the amount of avenin, a similar protein to gluten, is known to be much lower than the quantity of gluten in grains such as rye, barley, or wheat [66]. However, similar to other gluten-free grains, cross-contamination with other grains that contain gluten during the processing, handling, and transportation stages are a common occurrence since the equipment for processing as well as the farmlands are shared between these types of grains.

### 3.4. The Delicate Issue of Allergies to Gluten-Free Foodstuffs

Studies have shown that some gluten-free foodstuffs can produce serious allergic reactions and anaphylaxis [67,68,69]. A good example is buckwheat which is known to cause an IgE-mediated allergy [67]. For instance, common and Tartary buckwheat are known to possess several allergenic proteins such as Fag e 1, Fag e 2, Fag e 3, Fag t 1, Fag t 2, and Fag t 3, respectively [67]. Recently, three cases of food anaphylaxis to quinoa were also reported [68]. Prick tests and specific IgE allergies to quinoa were positive in the three cases. Another case of millet allergy that developed into wheat-induced anaphylaxis by cross-reaction between millet and wheat antigens was documented by Kotachi et al. [69].

### 3.5. High Cost of Gluten-Free Products

In a study conducted by Fry et al. [54], they reported that gluten-free products were 159 % higher in price compared to regular gluten-containing food products. This finding was in line with other studies which stated that gluten-free products had higher prices compared to gluten-containing products [70]. One of the reasons that contributes to the high price of gluten-free products is the fact that some countries, such as Mexico, do not provide subsidies for gluten-free food products [71]. Besides that, according to an online website, another reason that contributes to the high price of gluten-free products in the market is the complexity of the production process. For instance, the packaging, delivery, and production process requires stringent monitoring of the raw materials to assess gluten and other additional allergen contents and the use of raw materials (mainly corn and rice) that are usually more costly than wheat. An article by Lee et al. [72] mentioned that the cost burden of gluten-free products which are normally more expensive than regular gluten-containing products was one of the reasons that led to the tough adherence of some consumers to the gluten-free diet, especially for those who have symptoms of gluten sensitivity or celiac disease.

## 4. Conclusions and Future Trends

Individuals may restrict gluten from their diet for different reasons such as the improvement of gastrointestinal and non-gastrointestinal symptoms, as well as the belief that restricting gluten from the diet represents a healthy lifestyle. However, evidence has shown that gluten avoidance may be beneficial to some patients with gastrointestinal symptoms similar to those found with NCGS. However, high quality evidence supporting gluten avoidance for the symptoms of NCGS is not convincing. Therefore, there is a need for more etiological studies on NCGS. There are many potential health benefits of consuming gluten-free cereal products and beverages made from cereals or pseudo-cereals. Cereals and pseudo-cereals such as millet, sorghum, teff, quinoa, and buckwheat have the potential to increase the nutritional components and health benefits of products such as pasta, bread, cookies, crackers, and other unmentioned products that utilize gluten-free grains as their raw ingredients. The nutritional benefits of a gluten-free diet include having higher dietary fiber, micro-nutrients such as B vitamins and minerals, major nutrient components such as protein, and a higher antioxidant activity which could reduce other varieties of diseases such as cardiovascular disease, cancer, aging, and neuro-degenerative diseases. Moreover, some gluten-free foodstuffs can trigger serious allergic reactions and anaphylaxis. For instance, food anaphylaxis to buckwheat and quinoa has been reported. Besides that, micro-nutrients that are generally found in lesser amounts in the current commercialized gluten-free products can be solved by including flours from cereals and pseudo-cereals such as flours from millet, sorghum, teff, quinoa, and buckwheat. Therefore, it is recommended that food manufacturers should start replacing the existing market-available gluten-free products with those that have a greater health-promoting properties in order to address the needs of patients requiring nutrient supplementation.

## Figures and Tables

**Table 1 foods-10-02523-t001:** Summary of the nutrients and bio-active compounds and their benefits in selected gluten-free grains.

Grain Type	Nutrients and Bio-Active Compounds and Health Benefits	References
Sorghum	-rich source of some minerals (e.g., P, Mg, K, and Ca) and vitamin A. -contains low digestibility proteins which are probably caused by alpha-, gamma, and beta-kafirin proteins at the peripheral of the protein body which impedes digestion. -rich in fatty acids (linoleicacid, palmitic acid, stearic acid, and oleic) -rich in carotenoids, lycopene. -rich in phenolic compounds e.g., 3-deoxyanthocyanidins and tannins. -contains polycosanols which beneficially modulate in vitro parameters related to non-communicable diseases.	[17]
Millet (pear, finger, foxtail, kodo)	-contains protein ranging from 10 to 12.3 g/100 g. -their fiber ranges between 2 and 9 g. -iron 0.5–19.0 mg. -calcium is from 10 to 410 mg. -flavanoids are the main polyphenols (e.g., catechin, quercetin, luteolin, orientin, apigenin, isoorientin, vitexin, myricetin, isovitexin daidzein, and tricin). -The flavanoids exhibit a wide range of therapeutic properties; e.g., anti-inflammatory, anti-hypertensive, diuretic, analgesic, anti-cancer.	[18]
Teff	-protein content ranges between 12.8 and 20.9%. -contains a high concentration of insoluble polysaccharides. -lipid content is approximately 4.4%. -rich in flavones which are primarily C-glycosides such as apigenin-6,8-c-diglucoside, apigenin-8-c-glucosyl-7-*O*-glucosides, etc. -some of its health benefits include the prevention and treatment of celiac diseases, diabetes, and anemia.	[19]
Buckwheat	-rich in carbohydrates, protein, amino acid, fatty acid, vitamins (e.g., B-vitamins; B1, B2, and B3); minerals (e.g., Zn, Cu, Fe, mn, Se, K, Na, Ca, and Mg) and iminosugar. -the bio-active components are flavanoids (e.g., rutin, quercetin, orientin, isoorientin, vitexin, and isovitexin). -these bio-active compounds are known for their anti-inflammatory, antioxidant, anti-viral, and anti-ulcer properties.	[20]
Quinoa	-rich in protein with values in the range of 14–18%. -exhibits balanced amino acid composition. -rich in B-vitamins (B1, B2, and B3). -white, red, and black quinoa seeds are rich in phenolic especially flavonoids (e.g., protocatechinic acid, ferulic acid, quercetin, and kaempfeol derivatives, rutin). -rich in triterpene glycosides and they contain the aglycons of quercetin and isorhamnetin. -some of the health benefits of quinoa include anti-cancer properties and effectiveness against cardiovascular diseases.	[21]

**Table 2 foods-10-02523-t002:** Health benefits of gluten-free products made from gluten-free cereals.

Gluten-Free Products Made from Gluten-Free Cereals	Health Benefits
Bread	High antioxidant activity [33] Reduces blood pressure for people with mild hypertension [34] Lowers the blood glucose level [34] High in vitro antioxidant activity [35] Increased antioxidant activity [36]
Pasta	High nutritional contents such as dietary fiber, protein and polyphenols, and antioxidant activity [36] Low glycemic index, contains polyphenol and high resistant starch that contributes to a reduced rate of sugar absorption and LDL-cholesterol, and consequently lowers the risk of several cancers [37] Higher antioxidant potential than regular durum wheat pasta [30] Has good dietary fiber, protein, mineral, and contains phytonutrients [38]
Cookies	High dietary fiber, antioxidant activity, and essential amino acids [39] High source of antioxidant capacity, phenolic contents, and a low glycemic index [40]
Crackers	Higher antioxidant activity than control wheat crackers [41]
Non-alcoholic beverages	Has potential for anti-hypertensive and anti-diabetic effects [42] Reducing effects of DNA fragmentation, peroxidation of lipids, and protein oxidation [43] High antioxidative properties [44] *Kunnu* with tigernut extract has enhanced antioxidative potentials [45] Addition of lactic acid bacteria to beverage made from sorghum and amaranth composites displayed an increased digestibility of protein, increased bio-accessibility of zinc and iron and a higher availability of lysine [46]
Powdered drink mix	Has antioxidant properties and is low in calories [47]

## Data Availability

Not applicable.

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
