# Peer review of "Gluten-Free Cereal Products and Beverages: A Review of Their Health Benefits in the Last Five Years"

_foods, 2021, doi:10.3390/foods10112523_

Round 1

Reviewer 1 Report

Review comments for manuscript

 “Gluten-free cereal products and beverages: an overview of their health benefits»

               The manuscript “Gluten-free cereal products and beverages: an overview of their health benefits” technology” by Khairuddin & Lasekan is an interesting review paper, comprehensive and well written. In my opinion, it can to be published in Foods after minor revisions.

               For example,

  • the authors should check the whole text for repetitions (e.g. Lines 95-100) and punctuation mistakes,
  • the literature search plan and outcome with respect to chronological distribution and frequency of the publications should be provided in order to highlight the research, food industry and potential consumers’ interest,
  • “Future perspectives” section should be enriched.

Author Response

Review comments for manuscript

 “Gluten-free cereal products and beverages: an overview of their health benefits»

               The manuscript “Gluten-free cereal products and beverages: an overview of their health benefits” technology” by Khairuddin & Lasekan is an interesting review paper, comprehensive and well written. In my opinion, it can to be published in Foods after minor revisions.

               For example,

  • the authors should check the whole text for repetitions (e.g. Lines 95-100) and punctuation mistakes,DONE
  • the literature search plan and outcome with respect to chronological distribution and frequency of the publications should be provided in order to highlight the research, food industry and potential consumers’ interest,
  • “Future perspectives” section should be enriched.

The reviewer’s suggestion has been complied with

Reviewer 2 Report

The manuscript entitled “Gluten-free cereal products and beverages: an overview of their health benefits” submitted for revision in Foods had not been positively reviewed.

The topic is interesting and fits very well into the scope of Foods. It is a useful, factually sound but somewhat general overview.

There is a detailed but at the same time somewhat superficial introduction into the topic. This review reads more like a graduate dissertation and not a scientific review. The conclusion remains superficial and largely descriptive. 

There are many publications on gluten-free products. this publication adds nothing new. The nutritional and health properties of the products are not discussed in detail. It would be interesting to compile this information in the form of a table. Information on the nutrient content of individual cereals is lacking, and bioactive ingredients are not discussed.

The authors write (line 95-96):” Moreover the health benefits of cereals that are gluten-free such as sorghum, millet, and teff as well as pseudocereals like buckwheat and quinoa have been documented in many kinds of literature” but it is not stated in what publications, there is no reference to the literature. Please provide the publications on the basis of which these conclusions were drawn

Specific products and their technologies and how they affect them are not described. In the discussion, it is also worth considering various examples of gluten-free products described in other publications and their impact on health, ex.:

Regula, A. Cerba, J. Suliburska A. A. Tinkov. In Vitro Bioavailability of Calcium, Magnesium, Iron, Zinc, and Copper from Gluten-Free Breads Supplemented with Natural Additives. Biol. Trace Elem. Res. - 2018, vol. 181, 1, s. 140-146. doi: 10.1007/s12011-017-1065-4.
MichaÅ‚ Åšwieca, Julita ReguÅ‚a, Joanna Suliburska, Urszula ZÅ‚otek, Urszula Gawlik-Dziki. Effects of gluten-free breads, with varying functional supplements, on the biochemical parameters and antioxidant status of rat serum. Food Chemistry, 182, 1, 2015, 268–274
Guglielmetti, A., Fernandez-Gomez, B., Zeppa, G., del Castillo, D. (2019). Nutritional Quality, Potential Health Promoting Properties and Sensory Perception of an Improved Gluten Free Bread Formulation Containing Inulin, Rice Protein and Bioactive Compounds Extracted from Coffee Byproducts. Polish Journal of Food and Nutrition Sciences, 69(2), 157-166. https://doi.org/10.31883/pjfns-2019-0012
Allen, B., & Orfila, C. (2018). The Availability and Nutritional Adequacy of Gluten-Free Bread and Pasta. Nutrients, 10(10), 1370. https://doi.org/10.3390/nu10101370

Are the photos taken by the authors? I believe that photographs of cereals are not needed. They are illegible.

The information under the photos (lines 578, 593, 601) is illegible in the publication, which unfortunately does not allow for a proper assessment

Author Response

The manuscript entitled “Gluten-free cereal products and beverages: an overview of their health benefits” submitted for revision in Foods had not been positively reviewed.

The topic is interesting and fits very well into the scope of Foods. It is a useful, factually sound but somewhat general overview.

There is a detailed but at the same time somewhat superficial introduction into the topic. This review reads more like a graduate dissertation and not a scientific review. The conclusion remains superficial and largely descriptive. 

There are many publications on gluten-free products. this publication adds nothing new. The nutritional and health properties of the products are not discussed in detail. It would be interesting to compile this information in the form of a table. Information on the nutrient content of individual cereals is lacking, and bioactive ingredients are not discussed.

       The nutritional & health properties of selected grains have been compiled

        In the form of table. See Table1

The authors write (line 95-96):” Moreover the health benefits of cereals that are gluten-free such as sorghum, millet, and teff as well as pseudocereals like buckwheat and quinoa have been documented in many kinds of literature” but it is not stated in what publications, there is no reference to the literature. Please provide the publications on the basis of which these conclusions were drawn . This sentence has been deleted from this section.

Specific products and their technologies and how they affect them are not described. In the discussion, it is also worth considering various examples of gluten-free products described in other publications and their impact on health, ex.:

Regula, A. Cerba, J. Suliburska A. A. Tinkov. In Vitro Bioavailability of Calcium, Magnesium, Iron, Zinc, and Copper from Gluten-Free Breads Supplemented with Natural Additives. Biol. Trace Elem. Res. - 2018, vol. 181, 1, s. 140-146. doi: 10.1007/s12011-017-1065-4.
MichaÅ‚ Åšwieca, Julita ReguÅ‚a, Joanna Suliburska, Urszula ZÅ‚otek, Urszula Gawlik-Dziki. Effects of gluten-free breads, with varying functional supplements, on the biochemical parameters and antioxidant status of rat serum. Food Chemistry, 182, 1, 2015, 268–274
Guglielmetti, A., Fernandez-Gomez, B., Zeppa, G., del Castillo, D. (2019). Nutritional Quality, Potential Health Promoting Properties and Sensory Perception of an Improved Gluten Free Bread Formulation Containing Inulin, Rice Protein and Bioactive Compounds Extracted from Coffee Byproducts. Polish Journal of Food and Nutrition Sciences, 69(2), 157-166. https://doi.org/10.31883/pjfns-2019-0012
Allen, B., & Orfila, C. (2018). The Availability and Nutritional Adequacy of Gluten-Free Bread and Pasta. Nutrients, 10(10), 1370. https://doi.org/10.3390/nu10101370

         The suggestion is well taken and this has been addressed (Lines 448-

         455)

Are the photos taken by the authors? I believe that photographs of cereals are not needed. They are illegible.

        The photographs have been removed

The information under the photos (lines 578, 593, 601) is illegible in the publication, which unfortunately does not allow for a proper assessment

                   Photos have been removed

Reviewer 3 Report

The manuscript of Khairuddin & Lasekan focuses on review the health benefits of gluten-free cereal products. The objective is challenging due to the enormous amount of published articles on this argument. In addition, in the text, the authors treat the issue of consuming a gluten-free product diet too. Unfortunately, this generates high-flying expectations in the reader that are disappointed by the text.
At first, the authors must insert more updated citations. As an example, in the introduction, the so-called "non-celiac gluten sensitivity" lack a reference, while in recent years, other authors (e.g. Sergi et al. BMC Gastroenterol (2021) 21:5; Tanveer et al. J Coll Physicians Surg Pak. 2019) thoroughly revised the argument.
Moreover, in the introduction, the authors seem to forget to describe WDEIA. Under this point of view, the referee suggests citing the work of Morten et al. (Clinical and translational allergy 2019, Vol.9, 1). In this paper, the clinical threshold in wheat-dependent, exercise-induced anaphylaxis seems to be lowered in patients with a wheat-free diet. In contrast, the opposite is seen in patients on regular wheat intake. Therefore, a recommendation of wheat consumption is considered safe to these patients based on case history and challenge results. The authors must consider and discuss the drawbacks of a not fully motivated gluten-free diet in the present review.
In section 2.1. (Health benefits of gluten-free cereals and pseudocereals) it is not clear why the Authors omit to treat maize, rice, fonio, oats, etc. (see e. g. Current applications of gluten-free grains – a review. Woomer, Joseph S; Adedeji, Akinbode A Critical reviews in food science and nutrition, 2021, Vol.61; 1). 
In section 3. (Issues related to the consumption of gluten-free products), the authors omitted information about the delicate issue of allergy to gluten-free foodstuffs, e. g. the buckwheat based ones (see Norbäck, Dan; Wieslander, Gunilla, Plants 2021, Vol.10; 3).
In addition, the authors can find numerous other useful references for their review on diverse Foods Special Issues (e.g. "Improving the Sensory, Nutritional and Technological Profile of Conventional and Gluten-Free Pasta and Bakery Products").
As a consequence of the complexity of the proposed argument, it is suggested to limit the range of interest, for example, to the last five years literature or only to the use of a unique non-gluten-containing derivative. In addition, the authors should insert different tables to summarize the bibliographic references to any single argument.
Minor concerns are about few typographical mistakes (e. g. lines: 260, 404, 422, 444, 565).

Author Response

The manuscript of Khairuddin & Lasekan focuses on review the health benefits of gluten-free cereal products. The objective is challenging due to the enormous amount of published articles on this argument. In addition, in the text, the authors treat the issue of consuming a gluten-free product diet too. Unfortunately, this generates high-flying expectations in the reader that are disappointed by the text.
At first, the authors must insert more updated citations. As an example, in the introduction, the so-called "non-celiac gluten sensitivity" lack a reference,

      The reference is now provided (Line 69)

 while in recent years, other authors (e.g. Sergi et al. BMC Gastroenterol (2021) 21:5; Tanveer et al. J Coll Physicians Surg Pak. 2019) thoroughly revised the argument.

We quite agree with the reviewer on this and correction has

been effected with the statement in Lines 71-80

Moreover, in the introduction, the authors seem to forget to describe WDEIA. Under this point of view, the referee suggests citing the work of Morten et al. (Clinical and translational allergy 2019, Vol.9, 1). In this paper, the clinical threshold in wheat-dependent, exercise-induced anaphylaxis seems to be lowered in patients with a wheat-free diet. In contrast, the opposite is seen in patients on regular wheat intake. Therefore, a recommendation of wheat consumption is considered safe to these patients based on case history and challenge results. The authors must consider and discuss the drawbacks of a not fully motivated gluten-free diet in the present review.

Thank you. The issue of WDEIA has been included (Lines 91-95)

In section 2.1. (Health benefits of gluten-free cereals and pseudocereals) it is not clear why the Authors omit to treat maize, rice, fonio, oats, etc. (see e. g. Current applications of gluten-free grains – a review. Woomer, Joseph S; Adedeji, Akinbode A Critical reviews in food science and nutrition, 2021, Vol.61; 1). 

Thank you for your observation. We are quite aware of these other grains but we only mentioned some selected few.

In section 3. (Issues related to the consumption of gluten-free products), the authors omitted information about the delicate issue of allergy to gluten-free foodstuffs, e. g. the buckwheat based ones (see Norbäck, Dan; Wieslander, Gunilla, Plants 2021, Vol.10; 3).
In addition, the authors can find numerous other useful references for their review on diverse Foods Special Issues (e.g. "Improving the Sensory, Nutritional and Technological Profile of Conventional and Gluten-Free Pasta and Bakery Products").

Information about the delicate issue of allergy has been

Added (Lines 500-510)
As a consequence of the complexity of the proposed argument, it is suggested to limit the range of interest, for example, to the last five years literature or only to the use of a unique non-gluten-containing derivative. In addition, the authors should insert different tables to summarize the bibliographic references to any single argument.

Thank you for your suggestion. We have now limited our

range of interest to the last five years
Minor concerns are about few typographical mistakes (e. g. lines: 260, 404, 422, 444, 565).

          Corrections done

Round 2

Reviewer 2 Report

The publication still requires some minor corrections

Table 1 needs to be completed 

Sorghum - The author writes: “ rich source of some minerals & vitamins, -contains low digestibility proteins, -rich in unsaturated lipids. What vitamins? What minerals? What proteins, lipids?

The same with teff. What macro-and micro nutrients?

Buckwheat: What does it mean: „these bio-active compounds are known for their”?

Quinoa: -rich in vitamins? What vitamins?

Author Response

            Reviewer 2

The publication still requires some minor corrections

Table 1 needs to be completed 

Sorghum - The author writes: “ rich source of some minerals & vitamins, -contains low digestibility proteins, -rich in unsaturated lipids. What vitamins? What minerals? What proteins, lipids? [The corrections have been effected in Table 1]

The same with teff. What macro-and micro nutrients? [Macro-nutrients are protein, carbohydrate etc. However, this statement has been deleted since protein etc. Has already been mentioned in the table]

Buckwheat: What does it mean: „these bio-active compounds are known for their”?

       [these bio-active compounds are known for their anti-inflammatory, anti-oxidant, anti-viral & anti-ulcer properties ]

Quinoa: -rich in vitamins? What vitamins? [The B-vitamins. See Table 1]

Reviewer 3 Report

Although more similar to an excellent introduction to a PhD dissertation than a high-level scientific communication. However, I suggest changing the title to make it more appropriate to the content of the text, which is not just an overview of the recent literature on the "health benefits" of the gluten-free diet.

There are also still many typographical errors.

  • line 80 "oli-go"?
  • The format of the paragraphs changes continuously in the text. Compare lines 125 and 126 (where a strange graphic symbol also appears) or 135 and 137.
  • The use of commas in citations changes constantly. Compare, for example, line 144 (Gobbetti et al., [7]) with line 173 (Roa et al. [16]), 215, 245, 247, 294, 298, 302...
  • Line 189: there is a double square bracket.
  • In line 202 the spaces are omitted between squares (compared to line 205).
  • Line 206: a space must be removed.
  • "polyphenols" or "poly phenols" please make the text consistent.
  • Line 351: it is necessary to standardize the character.
  • Line 547: there are two commas.

The formatting of the bibliography should also be revised:

line 619 is missing a space; in line 661 there is a double space; I don't know if the use of points (...) in the bibliography (citations 30, 31, 33, 63, 64...) is compliant with editorial standards.

All of these changes (and undoubtedly many others I haven't seen) require the skill of an expert proofreader.

Author Response

         Reviewer 3

Although more similar to an excellent introduction to a PhD dissertation than a high-level scientific communication. However, I suggest changing the title to make it more appropriate to the content of the text, which is not just an overview of the recent literature on the "health benefits" of the gluten-free diet.

There are also still many typographical errors.

  • line 80 "oli-go"?[its oligo-saccharides]
  • The format of the paragraphs changes continuously in the text. Compare lines 125 and 126 (where a strange graphic symbol also appears) or 135 and 137.[The correction has been carried out]
  • The use of commas in citations changes constantly. Compare, for example, line 144 (Gobbetti et al., [7]) with line 173 (Roa et al. [16]), 215, 245, 247, 294, 298, 302…[Necessary corrections have been effected]
  • Line 189: there is a double square bracket.[REMOVED]
  • In line 202 the spaces are omitted between squares (compared to line 205).[Correction effected]
  • Line 206: a space must be removed.[Done]
  • "polyphenols" or "poly phenols" please make the text consistent.[Done]
  • Line 351: it is necessary to standardize the character.[Done]
  • Line 547: there are two commas.[One of the commas has been removed]

The formatting of the bibliography should also be revised:

line 619 is missing a space; in line 661 there is a double space; [Corrected]

I don't know if the use of points (...) in the bibliography (citations 30, 31, 33, 63, 64...) is compliant with editorial standards. [The points have been deleted]
